# The Impact of Low Cardiac Output on Propofol Pharmacokinetics across Age Groups—An Investigation Using Physiologically Based Pharmacokinetic Modelling

**DOI:** 10.3390/pharmaceutics14091957

**Published:** 2022-09-16

**Authors:** Karel Allegaert, Mohammad Yaseen Abbasi, Robin Michelet, Olusola Olafuyi

**Affiliations:** 1Department of Pharmaceutical and Pharmacological Sciences, KU Leuven, 3000 Leuven, Belgium; 2Department of Development and Regeneration, KU Leuven, 3000 Leuven, Belgium; 3Leuven Child and Youth Institute, KU Leuven, 3000 Leuven, Belgium; 4Department of Hospital Pharmacy, Erasmus Medical Center, 3015 GD Rotterdam, The Netherlands; 5Division of Clinical Pharmacology, Department of Medicine, Indiana University School of Medicine, Indianapolis, IN 46202, USA; 6Department of Clinical Pharmacy and Biochemistry, Institute of Pharmacy, Freie Universitaet Berlin, 12169 Berlin, Germany; 7Division of Physiology, Pharmacology and Neurosciences, School of Life Sciences, University of Nottingham, Nottingham NGT 2TQ, UK

**Keywords:** physiologically based pharmacokinetic modelling, propofol, low cardiac output, pharmacokinetics, neonate, developmental pharmacology, asphyxia, hypothermia, pediatrics, pharmacokinetics

## Abstract

Background: pathophysiological changes such as low cardiac output (LCO) impact pharmacokinetics, but its extent may be different throughout pediatrics compared to adults. Physiologically based pharmacokinetic (PBPK) modelling enables further exploration. Methods: A validated propofol model was used to simulate the impact of LCO on propofol clearance across age groups using the PBPK platform, Simcyp^®^ (version 19). The hepatic and renal extraction ratio of propofol was then determined in all age groups. Subsequently, manual infusion dose explorations were conducted under LCO conditions, targeting a 3 µg/mL (80–125%) propofol concentration range. Results: Both hepatic and renal extraction ratios increased from neonates, infants, children to adolescents and adults. The relative change in clearance following CO reductions increased with age, with the least impact of LCO in neonates. The predicted concentration remained within the 3 µg/mL (80–125%) range under normal CO and LCO (up to 30%) conditions in all age groups. When CO was reduced by 40–50%, a dose reduction of 15% is warranted in neonates, infants and children, and 25% in adolescents and adults. Conclusions: PBPK-driven, the impact of reduced CO on propofol clearance is predicted to be age-dependent, and proportionally greater in adults. Consequently, age group-specific dose reductions for propofol are required in LCO conditions.

## 1. Introduction

Children are not merely small adults as they undergo non-linear developmental changes, impacting pharmacokinetics [1]. These changes include development of organ system functions, maturation of cardiac output, organ perfusion and permeability, or glomerular filtration rate. Similarly, the ontogeny of cytochrome p450 (CYP) and non-CYP enzymes may result in differences in metabolic clearance [1,2]. Maturational and pathophysiological changes in children often co-exist, and this complicates drug treatment strategies. Though dosing design strategies commonly consider maturational changes occurring with age, there is usually less consideration on the impact of pathophysiology in patients and its effects on drug disposition [1,2]. Cardiac output (CO) is one of these pathophysiological changes.

In neonates, asphyxia and therapeutic hypothermia (TH) can reduce CO up to 33%, which is a known example of a disease-related impact on maturational pharmacokinetics [3,4,5]. Associated with this setting, the hepatic blood flow, and intestinal and renal blood flow are affected by CO reduction [6]. The same holds true for children with chronic heart failure, as illustrated for, e.g., carvedilol pharmacokinetics [7]. Based on observations in adults, it is, therefore, reasonable to expect a further decrease in clearance of high extraction drugs, with a pattern in neonates or children, co-modulated by physiology- and pathophysiology-related effects [7].

The clearance of propofol—a high extraction drug—is sensitive to CO changes in adults. It is a fast-acting drug used in general anesthesia for induction and maintenance of sedation in various invasive procedures. It is commonly used in neonates, children, and adults. Propofol is administered intravenously and has a rapid distribution with a large volume of distribution, rapid clearance, and high protein binding [8,9,10]. Uridine 5’-diphosphate-glucuronosyltransferase (UGT1A9) is the main metabolic pathway involved in propofol metabolic clearance, accounting for about two-thirds of total clearance. The remaining one-third is by hydroxylation, involving mainly CYP2B6 and a minor contribution of CYP2C9 [10,11]. Propofol clearance also involves significant extra-hepatic metabolic clearance, accounting for about 40% of its clearance, mainly driven by UGT1A9 [8,10]. The maturation of activity of UGT1A9, CYP2B6, and CYP2C9 evolves over the first weeks of life and beyond [12]. For example, the CYP2B6 activity in infants and younger children is said to be 1% and 50% of adult levels, respectively [13]. The hepatic abundance of UGT1A9 is thought to increase with age, as neonates and infants express 3% and 27% of adult UGT1A9 protein abundance levels, and 50% of ugt1a9 adult abundance protein levels reached at eight years [12,14].

The fact that propofol pharmacokinetics are sensitive to clearance-altering parameters, such as CO reduction and enzyme ontogeny, necessitates a pragmatic approach to assess the impact of the changes resulting from both pathophysiology and physiology throughout life, including in neonates and infants. Physiologically based pharmacokinetic (PBPK) modeling is non-invasive and, when implemented appropriately, is a reliable approach that may be used to make such assessments. PBPK models can incorporate both maturational and non-maturational features in pharmacokinetics analysis, enabling the assessment of the impact of (patho-)physiology on drug-specific pharmacokinetics [15,16,17]. PBPK models have been successfully used for similar analyses where, for example, the non-maturational impact of CYP2B6 genetic polymorphism and drug–drug interaction between efavirenz and lumefantrine alongside maturational changes in children was quantified [18]. More recently, Olusola et al. demonstrated that the reduction in CO, as a non-maturational parameter, did not significantly alter the acetaminophen pharmacokinetics in preterm neonates owing to immaturity of acetaminophen clearance pathways in preterms and the inherent low hepatic clearance capacity of acetaminophen [19].

Since propofol is classified as a high extraction drug, with clearance therefore expected to be altered upon CO reduction in adults, it serves as a good model compound to explore the impact of both physiology and pathophysiology on its pharmacokinetics across age groups [5,7,20]. The aim of this study was therefore to (i) assess the impact of reduced CO, such as asphyxia and TH in neonates, on propofol clearance capacity and (ii) how this affects safe and therapeutic concentration attainment across age groups, to explore dosing optimization strategies under low CO conditions. We hereby decided to explore a CO reduction range of −20% to −50%, as this is the clinically relevant range. Less CO reduction is difficult to discriminate from the normal variability, while a reduction of >50% in CO is a contra-indication for propofol use.

## 2. Materials and Methods

Simcyp^®^ (Simcyp^®^ Ltd., Certara, Sheffield, UK, Version 19) was used to predict propofol pharmacokinetics. This simulator has pre-validated virtual adult and pediatric population groups based on public health databases such as the US National health and Nutrition Examination Survey [21]. These virtual populations have similar interindividual variability in their demographic and physiological parameters as their real-world counterparts and can thus be used for population simulations. Population sizes for simulations included a 20 × 10 trial design with 200 subjects.

A previously developed, validated and peer-reviewed propofol model (Michelet et al.) was optimized [22]. Details of the optimization process and final optimized parameters are provided in the Appendix A [15,22,23,24,25,26]. After optimization, the propofol model was validated with clinical data in adults and children (all age subcategories) retrieved from published literature, also provided in the Appendix A [27,28,29,30,31,32,33,34,35,36,37,38].

### 2.1. Hepatic and Renal Extraction Ratio Determination across Age Groups

Following validation of the optimized propofol model, the Morse et al. dosing model [39] was simulated in neonates, infants, children, and adolescents, while the Roberts et al. dosing model [40] was implemented in adults to predict organ clearance, required to determine the extraction ratio of propofol.

The hepatic and renal extraction ratios of propofol were subsequently calculated from the predicted hepatic clearance (*CL_H_*) and metabolic renal clearance (*CL_MR_*), respectively, and their respective organ blood flows, that is, the hepatic blood flow (*Q_H_*) and the renal blood flow (*Q_R_*), using Equations (1) and (2). A well-stirred model was assumed. *Q_H_* was determined by multiplying CO by the fraction of cardiac output that perfuses the liver from the arterial and portal supply. Similarly, *Q_R_* was determined by multiplying CO by the fraction of cardiac output that perfuses the kidney.
(1)$$Hepatic\;extraction\;ratio\;\left(E_{H}\right)=\frac{CL_{H}}{Q_{H}}\;$$
(2)$$Renal\;extraction\;ratio\;\left(E_{R}\right)=\frac{CL_{MR}}{Q_{R}}$$

### 2.2. Impact of CO Reduction on Systemic Propofol Clearance across Age Groups

Using the manual infusion dosing strategies in the Morse and Robert models [39,40], the impact of CO reduction on systemic propofol clearance was determined by calculating the percentage relative change under reduced CO conditions. To mimic the potential impact of CO reduction, a 20, 30, 40 and 50% CO reduction was implemented. This was achieved by running the simulations after decreasing the CO input within the simulator by the respective percentage reductions. For example, in the case of a 20% CO reduction, within the simulator, the CO input code was multiplied by 0.8. The resultant systemic clearance was retrieved. The retrieved clearance under each reduced CO condition and under the normal CO condition was used to determine the effect of reduced CO on propofol clearance using Equation (3).
(3)$$Relative\;Clearance\;\left(CL\right)\;change\;\;\left(\%\right)=\frac{CL\;under\;normal\;CO-CL\;under\;reduced\;CO}{CL\;under\;normal\;CO}\times 100\;$$

### 2.3. Impact of Reduced Cardiac Output on Attainment of Target Propofol Concentrations

The Morse and Robert models are expected to achieve a target plasma concentration of 3 µg/mL (80–125%) in children [39] and adults [40], respectively. According to Morse et al. and Robert et al., this target is expected to achieve anesthesia while avoiding side effects from high levels of propofol in the plasma, such as hypotension. Using these models, concentrations achieved 2 h after start of manual infusion dosing under normal CO conditions and 20, 30, 40 and 50%-reduced CO conditions were determined [39,40].

### 2.4. Dose Reduction Exploration to Achieve Target Concentrations in Reduced Cardiac Output Conditions

A dose optimization exploration was conducted under reduced CO conditions. The dosing optimization strategy involved the percentage total dose reduction. The percentage dose reduction was iteratively implemented until the predicted concentrations under reduced CO conditions achieved the target concentration range.

## 3. Results

### 3.1. Hepatic and Renal Extraction Ratios across Age Groups

Figure 1 shows that E_H_ and E_R_ increase from neonates until adulthood. The mean E_H_ was borderline intermediate extraction (0.34) in neonates until adolescence, where the mean E_H_ increased to high extraction (0.74) (Figure 1). The predicted mean E_R_ followed the same pattern with age, with a low extraction range at 0.02 in neonates until infancy, where E_R_ increases to intermediate extraction at 0.34 until adulthood (0.55) (Figure 1).

### 3.2. The Impact of Reduced Cardiac Output on Systemic Propofol Clearance across Age Groups

The absolute clearance of propofol increased with age (Figure 2; Table 1). Similarly, the impact of CO reductions on clearance was lowest in neonates as the relative change in clearance following all CO reductions increased with age (Figure 2; Table 1). The greatest difference in relative clearance changes between age groups occurred between neonates and infants, suggesting the greatest developmental changes impacting propofol clearance in the first year of life. Figure 2 shows that higher-percentage CO reductions resulted in greater relative percentage reductions. Across age groups, the relative change in clearance was greater under higher CO reduction scenarios compared to lower CO reduction scenarios.

### 3.3. Impact of Reduced Cardiac Output on Attainment of Target Propofol Concentrations

Under normal CO conditions, predictions fell within the target concentration at 2 h of infusion to safely maintain anesthesia (3, 2.4–3.75 µg/mL) [39]. All reduced CO conditions resulted in raised plasma concentrations at 2 h after infusion in all age groups; however, under 20 and 30%-reduced CO conditions, the predicted mean plasma concentrations remained within 80 and 125%, but were raised above the upper limit (125%) under 40 and 30%-reduced CO conditions in all age groups (Figure 3).

### 3.4. Dose Reduction Exploration to Achieve Target Concentrations in Reduced Cardiac Output Conditions

Under 40 and 50%-reduced CO conditions, the percentage of virtual subjects achieving target plasma concentration ranged between 7–59% across all age groups (Figure 4). A 15% reduction in the total dose administered in neonates, infants and children and a 25% dose reduction in the total dose administered in adolescents and adults (see Appendix A, for actual dosing schedule) resulted in an increase in the number of subjects achieving the target plasma concentration across all age groups, with up to 89% of the virtual population achieving the target concentration range in adults with 40%-reduced CO. (Figure 4).

## 4. Discussion

PBPK models have become increasingly important in assessing pharmacokinetics in special populations with complex pathophysiological conditions [15,16,17]. This is particularly important in pediatrics, whose immature physiology may already imply altered pharmacokinetic properties compared to adults. Treatment strategies may be further complicated by pathophysiologic events, such as low CO [16]. Clinically, children are exposed to propofol in low CO conditions, such as neonatal asphyxia and TH, after cardiac surgery or heart failure [5,6,7]. The effect of reduced CO on propofol pharmacokinetics has been described in adults [20], while the combined effect of age and low CO (−20 to −50% reduction) has—until now—not yet been explored.

The propofol models used in this analysis were optimized from a published model (Michelet et al.) [22] satisfactorily recovered pharmacokinetic profiles and estimated clearance parameters reported for children and adults (Appendix A) [27,28,29,30,31,32,33,34,35,36,37,38]. In adults, total propofol clearance has been reported with some variability, with a clearance of 1.54, 2.2 or 2.64 L/min [20,41,42]. This is greater than hepatic perfusion (0.8–1.2 L/min) [43] and similar to the model-predicted clearance in this analysis implementing the Robert et al. model [40] (Table 1). In children, applying the Morse et al. model [39], the model-predicted systemic clearance ranged from 0.048 ± 0.02 to 1.58 ± 0.5 L/min between neonatal life to adolescence (Appendix A), similar to literature-reported values across the pediatric-age group (0.034 to 1.1 L/min) [32,33,34,35].

The E_H_ and E_R_ ratios of propofol were reported to be 0.87 ± 0.09 and 0.7 ± 0.13, respectively, in adults [6]. They were predicted to be 0.75 ± 0.2 and 0.55 ± 0.22 in adults in the current study (Figure 1). As the renal pathway constitutes a significant proportion of propofol clearance, it is also important to consider the impact of low CO on renal extraction of propofol [8]. The effect of age on the E_H_ of drugs has been previously reported when Salem et al. showed that the E_H_ may increase with age [44]. This age-dependent alteration was attributed to the disproportionate maturation of E_H_-impacting physiological and biochemical parameters occurring after birth, such as maturation of the fraction of unbound drugs, intrinsic clearance of unbound drugs, or hepatic blood flow [44]. A similar principle can be expected to impact the metabolic clearance occurring in the kidneys and, therefore, the propofol E_R_. The extent to which either of these parameters impact the extraction rate depends on the susceptibility of the drug clearance to alteration of given parameters. For example, Takizawa et al. showed that—despite high protein binding of propofol—hypoalbuminemia did not significantly alter its E_H_ [45]. In contrast, CO is documented to affect propofol clearance in adults [3,6,10].

Age-dependent alterations in propofol E_H_ and E_R_ were not yet explored. The current study demonstrates that both E_H_ and E_R_ increase with age: neonates have an E_H_ within the intermediate extraction range (0.34 ± 0.18), which evolves into high extraction (0.75 ± 0.2) in adults. Related to E_R_, neonates have a low extraction range (0.12 ± 0.06), which evolves into intermediate extraction (0.55 ± 0.2) in adults (Figure 2). This observation is similar to that observed with midazolam, with an E_H_ low extraction (0.02) pattern at birth and intermediate extraction (6.0) during adulthood [44].

Non-maturational parameters can also impact clearance and extraction ratios. For instance, CO changes and the resultant reduction in organ perfusion may influence the extraction capacity of such organs if the drug behaves as a high extraction compound. The impact of various degrees of CO reduction observed in this current study showed that the proportional impact of CO on clearance depends on age, with a lower impact in neonates and young infants compared to adults (Figure 3). This re-illustrates the fact that a neonate is not a small adult [1]. Furthermore, an age-dependent proportional impact of the severity in CO reduction on propofol clearance was observed (Figure 3). This is consistent with the fact that propofol possess some level of extraction capacity in early life, and this capacity becomes greater with age (Figure 1).

The clinical impact of CO on propofol PK is debatable. Leslie et al. 1995 demonstrated a significant increase in propofol plasma concentrations in patients with mild hypothermia [2] and, since reduced CO is a significant feature in hypothermic conditions [5], there is the expectation that propofol-altered PK in hypothermic patients is likely due to reduced CO. Though Bienert et al. 2020 did not identify clinically measured CO reduction as a significant covariate which altered the variability of propofol PK in their study patients (with only about 20% alteration to propofol variability when CO was included in their model), they found significant model-predicted CO correlations between CO and propofol PK. Bienert et al. 2020 noted that their clinically measured CO may not have significantly improved variability in propofol PK parameters because of the manner in which CO was measured in their study [20].

The pathophysiological effect of low CO, as observed in clinical conditions, such as neonatal patients with asphyxia undergoing TH or pediatric patients with heart failure, on propofol clearance may require considerations on how this impacts propofol dosing. In neonates, propofol is an intermediate extraction drug and reduced CO alters clearance, though the extent of change increases with age. Being aware that several dosing models are available, such as the Eleveld [46] and McFarlan [47] models, the Morse model was applied in this current study, and a target propofol plasma concentration of 3 (80–125%) µg/mL [39]. Applying this target, CO reduction (up to 30%) did not move the mean plasma concentrations outside of this target range across all age groups (Figure 3).

However, further-reduced CO (by 40 or 50%) resulted in mean plasma concentrations above the upper limit of the target concentration (Figure 3), which increases the risk of side effects associated with raised propofol concentrations such as hypotension [39]. This is consistent with Leslie et al. 1995, which reported that in hypothermic (34 °C) subjects, the propofol concentrations were significantly raised compared to normothermic (37 °C) subjects [3]. However, as this study did not measure the CO in the hypothermic subjects, it is difficult to determine to what extent CO reduction in these patients resulted in altered clearance and, subsequently, raised propofol clearance [3].

The impact of up to 50% CO reduction on the achieved plasma concentration is particularly crucial in neonates undergoing TH, as right- and left-ventricular output in asphyxiated TH cases are 108 (−51%) and 107 (−52%), significantly lower when compared to normative (224 and 222 mL/kg/min) values [48]. Along the same line, a 33% CO reduction has been documented in TH patients in a paired study design (169 versus 254 mL/kg/min) [5]. A 15% total dose reduction in neonates, infants and children under 40% and 50%-reduced CO conditions increased the number of virtual subjects achieving the expected target concentration, from as low as 26% to up to more than 80% achieving the target concentration range following dosing optimization (Figure 4). In adolescents and adults, a 25% total dose reduction resulted in up to 89% of virtual subjects achieving the target concentration range, up from as low as 7% achieving the target concentration range under reduced CO conditions (Figure 4). The greater amount of dose reduction in adults and adolescents compared to the younger age groups reflects the greater magnitude of the effect of CO on clearance in adults and adolescents compared to younger children.

While the result from this study provides valuable insight into the impact of reduced CO on clearance and dosing strategies of propofol across age groups, a limitation of the study was the inability to validate the model in patients with specified reduced CO. Though there are published studies showing the impact of reduced CO on propofol pharmacokinetics, the dosing methods used in these studies were target concentrate-based, making it impossible to simulate a particular dosing strategy, as one would expect for manual infusions [20,49]. However, it does support the practice to be cautious with propofol in low CO patients and this makes the collection of pharmacokinetic samples in pediatric studies even more challenging. Furthermore, our findings could provide a framework for future clinical studies where the effect of CO on propofol PK, including dosing optimization strategies, could be studied in children.

In conclusion, we have shown that—using a PBPK modelling approach—that the hepatic and renal extraction of propofol is predicted to change with age and the impact of reduced CO on propofol clearance is age-dependent, with the greater impact in adults. In addition, age group-specific dosing optimizations are required in low CO conditions.

## Figures and Tables

**Figure 1 pharmaceutics-14-01957-f001:**
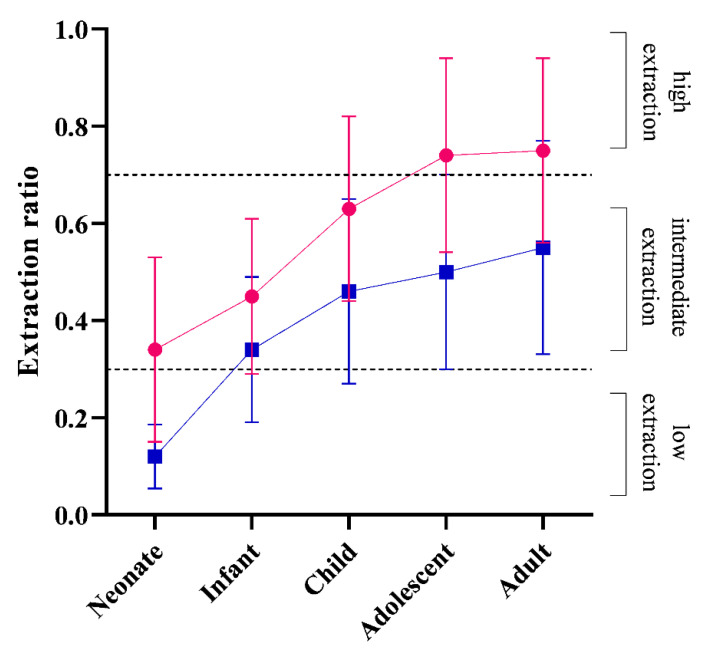
Predicted changes in hepatic extraction ratio (E_H_) and renal extraction ratio (E_R_) with age. Red circle and line: E_H_; blue circle and line: E_R_; black broken lines: low extraction ratio target (0.3) and high extraction ratio target (0.7). Data points and error bars represent mean and standard deviations of predicted extraction ratios, respectively.

**Figure 2 pharmaceutics-14-01957-f002:**
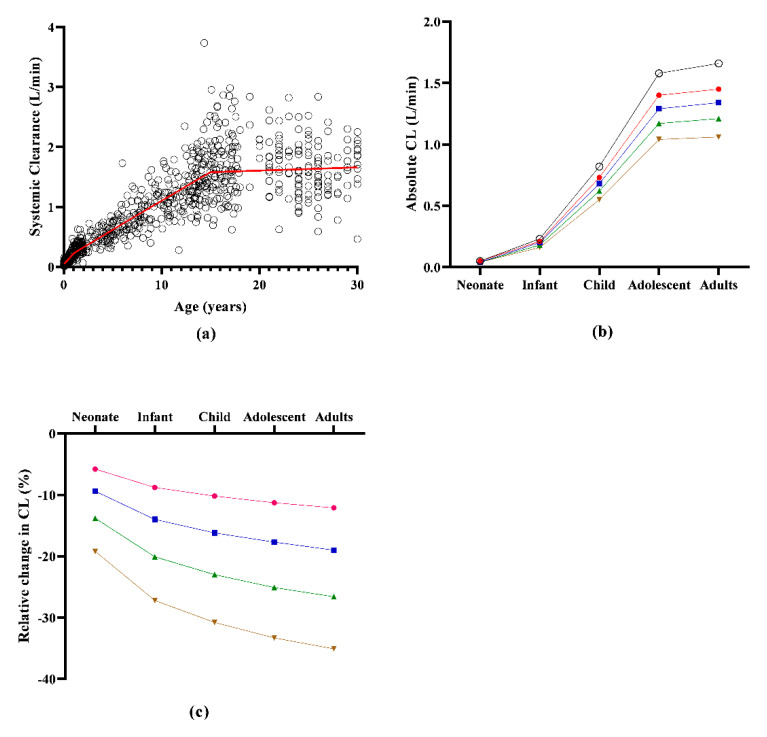
Predicted changes in systemic clearance of propofol under normal or reduced CO conditions. (**a**) Predicted changes in absolute clearance with age. (**b**) Predicted changes in absolute clearance with age under reduced CO conditions. (**c**) Relative percentage changes in systemic clearance under reduced CO conditions with age. Open circles represent individual clearance in virtual subjects. (**b**) Absolute clearance and (**c**) relative change in clearance of propofol under reduced cardiac output conditions across age groups. For all Figure 2a–c, black open circle and line, red circles and line, blue squares and line, green triangle and line and brown inverted triangles and line represent clearance under normal, 20%, 30%, 40% and 50%-reduced cardiac reduction, respectively.

**Figure 3 pharmaceutics-14-01957-f003:**
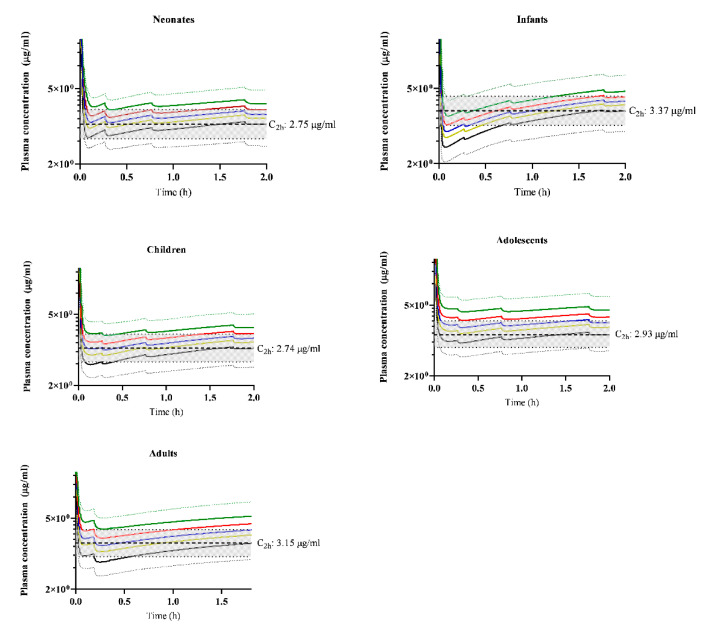
Simulated propofol plasma concentrations under various cardiac output conditions. Solid green, red, blue, yellow and black lines represent the predicted mean plasma propofol concentration under 50%, 40%, 30%, 20% and normal cardiac output (CO) conditions, respectively. Dotted green and black lines represent the predicted 95th and 5th percentile of the highest and least predicted mean concentration. The bold black dashed line is plasma concentration achieved at 2 h and the grey shaded area bordered by a thin black dashed line is 80–125% of target plasma concentration achieved at 2 h.

**Figure 4 pharmaceutics-14-01957-f004:**
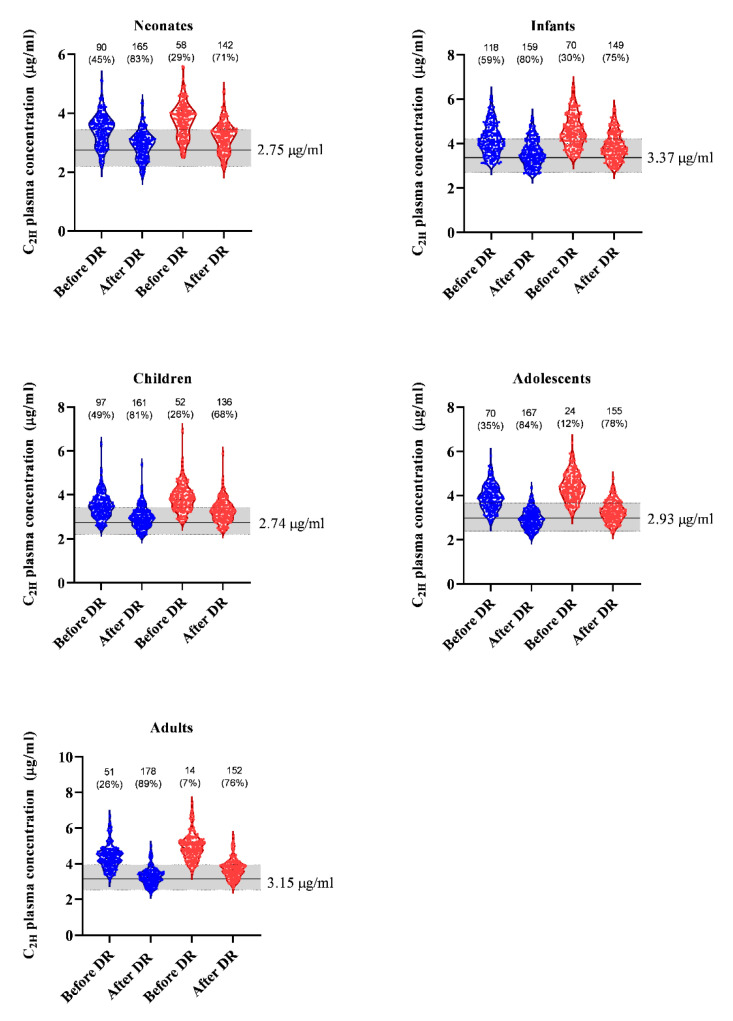
Predicted C_2H_ concentrations achieved following dosing optimization. Distribution of predicted individual plasma concentrations of propofol at 2 h after start of dosing under 40%-reduced CO conditions (blue plots) and 50%-reduced CO conditions (red plots). The black line is the mean concentration under normal CO conditions and the grey shaded area bordered by a thin black dashed line is the target concentration range. DR: dose reduction.

**Table 1 pharmaceutics-14-01957-t001:** Predicted clearance of propofol across age groups.

Cardiac Output	Neonates		Infants		Children		Adolescents		Adults	
	Absolute (L/min)	Relative (%)	Absolute (L/min)	Relative (%)	Absolute (L/min)	Relative (%)	Absolute (L/min)	Relative (%)	Absolute (L/min)	Relative (%)
Normal CO	0.05 (0.02)		0.23 (0.11)		0.82 (0.35)		1.58 (0.50)		1.66 (0.45)	
20% ↓CO	0.05 (0.02)	−5.8 (2.3)	0.21 (0.10)	−8.8 (2.4)	0.73 (0.30)	−10.2 (2.2)	1.40 (0.41)	−11.3 (2.2)	1.45 (0.36)	−12.1 (2.3)
30% ↓CO	0.04 (0.018)	−9.4 (3.6)	0.20 (0.09)	−14.0 (3.7)	0.68 (0.27)	−16.2 (3.3)	1.29 (0.37)	−17.7 (3.3)	1.34 (0.32)	−19.0 (3.3)
40% ↓CO	0.04 (0.016)	−13.8 (5.0)	0.18 (0.18)	−20.1 (4.9)	0.62 (0.24)	−23.0 (4.3)	1.17 (0.32)	−25.1 (4.2)	1.21 (0.28)	−26.6 (4.3)
50% ↓CO	0.04 (0.014)	−19.2 (6.4)	0.16 (0.07)	−27.2 (6.1)	0.55 (0.21)	−30.8 (5.3)	1.04 (0.27)	−33.3 (5.0)	1.061 (0.23)	−35.1 (5.1)

CO: cardiac output; relative values were calculated relative to normal CO (see Section 2.2).

## Data Availability

The data underlying the figures and the Simcyp output files are provided in the Appendix A; other information can be provided upon reasonable request by an e-mail to the corresponding author.

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
