# Peer review of "The Impact of Low Cardiac Output on Propofol Pharmacokinetics across Age Groups—An Investigation Using Physiologically Based Pharmacokinetic Modelling"

_pharmaceutics, 2022, doi:10.3390/pharmaceutics14091957_

Round 1
Reviewer 1 Report
Please find my review in the attached PDF.

Author Response
Reviewer 1
Within the manuscript
“The impact of low cardiac output on propofol pharmacokinetics across age groups - an investigation using physiologically based pharmacokinetic modelling.”
an existing model of propofol pharmacokinetics was applied to predict the effect of changes in cardiac output on propofol pharmacokinetic parameters such as extraction ratio and clearance.
Major
Comments 1: Data/code/model availability
- The model and model simulations cannot be evaluated without the availability of the model and access to the software and analysis code. Linking to the original model publication (Michelet2018) is not sufficient, because the model/model equations are not readily available in the original manuscript. In the current state none of the work/results can be checked, reproduced or evaluated. Providing a computational model/analysis in 2022 without access to the code/analysis/model is not sufficient.
For some discussion/background see
â—‹ Peng RD. Reproducible research in computational science. Science. 2011 Dec
2;334(6060):1226-7. doi: 10.1126/science.1213847. PMID: 22144613; PMCID: PMC3383002.
â—‹ Barnes N. Publish your computer code: it is good enough. Nature. 2010 Oct 14;467(7317):753.
doi: 10.1038/467753a. PMID: 20944687.
- Provide the model as a SIMCYP file in the supplement
- Provide documentation of the model equations and parameters used
- Provide all data in computer readable formats in the supplement. I.e. there should be a table/tables with all the data used for evaluation (digitized pharmacokinetic datasets with meta data). If possible upload the datasets to the pharmacokinetics database PB-PK (disclaimer: the reviewer is involved in this project, but can support the authors in sharing the digitized data via PK-DB).
- Provide simulation data/results shown in figures as tables in the supplements. The data shown in the figures should be provided in computer readable formats in the supplement (i.e. time course predictions & predictions of ER, changes in clearance with CO and so on).
Reply 1: The following reply addresses all the issues raised under the ‘data/code/model availability’ heading
We have carefully reflected on this request, but the authors respectfully disagree that the information requested above is necessary in this current research for the following reasons:
- The focus of this research was not to develop a model, but to use an already existing and previously validated model to predict a ‘what if’ scenario. So, we believe the better approach to presenting our findings was to focus more on the results involving the ‘what if’ predictions and not the model development.
- As explained in the manuscript (see P3L104-105), an existing propofol model was used in this research. The existing model was optimised, the process of optimisation and the final parameters of the optimisation process were detailed in the supplementary material (see P1L3-15 supplementary material). We also believe that the relevant data required for reproducibility of the model have been reported in Michelet et al paper [1] and in this current study.
- We believe that the codes and models used in the software (Simcyp) is both widely and reliably used in the pharmaceutical industry and drug regulatory organizations such as the FDA, for example [2, 3].
- All relevant equations and models (including population models) used in within Simcyp has been previously published; is publicly available and have been utilized successfully in several case studies [4-7]
Based on these explanations, the authors believe no further action is required to address the points above raised by the reviewer.
Major changes in figures/tables
Figure model overview
Comment 2:
- provide a figure model overview which depicts the model structure, organs, and information on the changes/differences of the different populations.
Reply 2: As explained in reply 1 above, the authors believe this is not necessary in this current study, since the focus of the research was not the development of a model. The model structure is commonly requested for popPK models, but are ‘standard’ for PBPK models, and look like the figure copied below (copyright protected).
Based on our experience, a figure is only added if e.g. a specific issue has been added, like eg placenta, but this is not the case in this analysis.
Figure 1
Comment 3:
- error bars missing from the simulations! Virtual populations are simulated resulting in distributions of pharmacokinetics parameters such as ER. The SD or similar error measurement of the population predictions must be added to the figures
Reply 3: We thank the reviewer for this suggestion, and the authors have included the SD (see Figure 1 – P5L158)
Comment 4:
- in addition, show ER changes in extraction ratio with CO (analogue to Figure 2). The focus of this work is the effect on CO on the pharmacokinetic parameters.
Reply 4: The authors do not believe this data adds more information to the analysis since the impact on CL was already simulated, and this a merged estimate, inclusion ER changes. We hope that the reviewer can agree on this.
Comment 5:
- don’t use marker and color to encode EH and ER, change in color is sufficient.
Reply 5: The authors have corrected this (see Figure 1 – P5L158)
Comment 6:
- don’t use colors for the dashed/dotted lines (make gray) and use the same linestyle (dashed) for the two separation lines (colors and different linestyles just add visual clutter)
Reply 6: The authors have corrected this (see Figure 1 – P5L158)
Figure 2
Comment 7:
- provide additional panels: 1. absolute clearance values; 3. absolute clearance ~ change in CO; 4. relative CL changes ~ changes in CO
Reply 7: The authors have added the information on these estimates (see Figure 2 – P7L176)
Comment 8:
- Probably Figure 1 & 2 could be combined consisting of multiple panels looking at the changes of pharmacokinetic parameters with CO.
Reply 8: The authors believe Figures 1 and 2 should remain separate because this makes the flow of result consistent with the methods and line of discussion. Merging both figures would not provide additional insights, but would result in a crowded figure, rather more difficult to interpret.
Comment 9:
- errorbars, prediction intervals missing, i.e. show the error bars corresponding to your variation of the model prediction
Reply 9: Along the same line, The authors believe inclusion of error bars to the current figure will make the figure too busy. We therefore have added an additional table which includes the means and SD values of absolute and relative values have been included (see Table 1 – P7L188)
Comment 10:
- Relative change is misleading and must be negative! I.e. a reduction in CO results in a reduction in clearance, but in the figure it looks like an increase. Change must be calculated relative to baseline i.e. the change must be calculated as
Relative clearance change [%] = (CL_0 - CL_reduced)/CL_0 * 100 [%]
This will give negative values and should be depicted on an y-axis which goes down. See
https://www.ahajournals.org/doi/10.1161/01.ATV.15.5.678 for an example.
This is also very confusing written in the text: p4L154: “In Figure 2, the relative change in clearance following CO reductions increased with age with the least impact of CO reduction in neonates across all CO reduction scenarios.”
This does not make clear that clearance was reduced in reduced CO, but sounds like an increase!
Reply 10: The authors do not believe ‘relative change’ is misleading. We argue that since we already expect that there would be a reduction in clearance with reduce CO, we were interested in understanding the magnitude or extent of change (reduction, in this case) as opposed to the direction of change. However, to avoid any confusion that may arise from the presentation of our findings, we have rescaled the y-axis to depict the negative values (see Figure 2 – P7L176) and the authors have reworded the phrase referred to above (see P7L164 – 166). This does not really alter the findings and the figures, but respects the request of the reviewer.
Comment 11:
- don’t use color and marker to encode the change (color is sufficient)
Reply 11: The authors have made this correction (see Figure 2 – P7L176)
Comment 12:
- add the 10% change
Reply: The authors do believe that the range of percentage reductions chosen in this research would provide a broad range of perspective on the effect of CO, and the impact is proportional between 20 and 50 % reduction. Also, the authors believe that a 10% reduction is CO is not clinically relevant and cannot be discriminated from the normal reference variability in estimates (eg cardiac ultrasound). We have included a statement to explain the reason for starting the percentage reduction at 20% (PLEASE address KAREL)
Comment 13:
- Overall validation data missing in these plots. Not a single data point included for propofol in Figure 1 and 2. What are reported extraction ratios? What are the reported clearance values? How is clearance affected by CO (look at subset of heart disease subjects); See supplementary table S5; Depict the data of Table S3 as validation data in the plot of clearance
Reply 13: The authors would like to draw the reviewer attention to supplementary material (see P2L32-39; Figure S1 and Table S3) were validation data were already reported. The relevant PK parameters (CL and t1/2) which were reported in the observed clinical data were reported (see Table S3). Extraction ratios are not routinely reported in the clinical studies; therefore, these were not tabulated. However, the similarity between the predicted extraction ratios in this current study and a published study were these were reported was commented on in the discussion section (see P13L269 – 271).
Validation was not possible in cardiac subjects because there are currently no published studies were cardiac output and propofol PK parameters was measured in patients using manual infusion dosing, especially in neonates or infants. This was stated as part of the limitation of the study in the manuscript (see P15L351 – 355)
Figure 3
Comment 14:
- the dashed curves are not helping much (remove from plot). The whole plot is too cluttered and small.
Reply 14: The authors have adjusted the plot (see Figure 3– P10L199)
Comment 15:
- order legend entries in order of curves i.e. the green entry on top (this makes it much easier to read the legend). Also only show the legend only once large (bottom right, no need for duplication which clutters the plot)
Reply 15: The authors have corrected this (see Figure 3– P10L199)
Comment 16:
- add 10% change to simulations and curves
Reply 16: The authors would refer the reviewer to reply 12
Comment 17:
- make normal curve black
Reply 17: The authors have made this correction. (see Figure 3– P10L199)
Figure 4
Comment 18:
- Figure 4 is basically a duplicate of Figure 3 => make a single figure; i.e. remove Figure 4 or move to supplement and provide better analysis/presentation of the CO changes/dose adaptation.
Reply 18: The authors have move this Figure 4 to the supplement and provided a new Figure 4 (see Figure 4 – P12L233; supplementary material Figure S2; P3L60)
Comment 19:
- This is not a good plot to evaluate the dosing adjustment. The plot can be moved to the supplement, but a more systematic analysis should be provided in the main text. E.g. barplot of percentage of population achieving TR before and after dosing adjustment. Or the distribution of distances in concentration from target concentration before and after dose adjustment.
Reply 19: The authors have included another Figure as advised (Figure 4 – P12L233)
Comment 20
- Same general changes as in Figure 3.
Reply 20: The authors refer the reviewer to the corresponding responses.
Comment 21
- Simulations of CO Reduction must be performed for the virtual populations (not only for mean model).
The results should be calculated on the virtual populations and mean +- SD results must be reported.
Reply 21: The authors would like the reviewer to note that the data presented were simulated in virtual populations. To address the reviewer’s comment, the SD have been included in data were necessary (see previous replies)
Comment 22:
- Also the achievement of therapeutic range (TR) must be evaluated for the populations. I.e. how much of the population [%] achieves TR before and after dosing adjustment. You are interested in optimizing the 2 hr concentration, so show the distribution of the two hour concentrations instead of/in addition to the time courses in Figure 4.
Reply 22: The authors have added an additional figure. (Figure 4 – P12L233)
Table 1
Comment 23:
- belongs in the supplement, does not provide any information for the main manuscript. The 15% and 25% are stated in the main text, no need to put the complete table in the main manuscript.
Reply 23: The authors have moved to this to the supplement (see supplementary material Table S4)
Figure S1
Comment 24
- This figure belongs in the main manuscript, not the supplement. This shows the model performance and is crucial for evaluation of the model and model predictions.
Reply 24: The authors believe that since the focus of the study was not model development, Figure S2 should remain within the supplementary material. In this way, it provide circumstantial information, but is not crucial to assess the paper in itself. We hope that the reviewer can agree with this approach.
Comment 25:
- unclear which data corresponds to which study; please provide the dosing schema with the figure; number of subjects should be stated. Clearly indicate which data belongs to which of the 5 groups (by color). See Rüdesheim2020 Figure 2 for how this should look; Also provide a nice table of the studies as seen in the Rüdesheim2020 supplement.
Rüdesheim S, Wojtyniak JG, Selzer D, Hanke N, Mahfoud F, Schwab M, Lehr T. Physiologically Based Pharmacokinetic Modeling of Metoprolol Enantiomers and α-Hydroxymetoprolol to Describe CYP2D6 Drug-Gene Interactions. Pharmaceutics. 2020 Dec 11;12(12):1200. doi: 10.3390/pharmaceutics12121200. PMID: 33322314; PMCID: PMC7763912.
Reply 25: The dosing scheme and studies used was already reported in Table S2. The authors believe it will be repetitive to include this again in the legend (see Table S2)
Table S3
Comment 26:
- Table S3 belongs in the main manuscript. These are a few data points which are predicted by the model.
Reply 26: The author responds to this comment in the same manner as reply 24. In addition, the authors have added an extra table in the main manuscript which include both absolute and relative values of propofol clearance. (see Table 1 – P11L189)
All figures
Comment 27:
- The legends are absolutely minimal at the moment and do not provide sufficient information to understand the plots (figure and legend should be self-explanatory without the need to read the manuscript). The information in the figure and what was simulated/how must be shortly described in the legends. E.g. the following text in the manuscript belongs in the figure 2 legend not in the main text:
p5L165
“Brown inverted triangles and line; green triangles and line; blue squares and line; red circles and line represent relative change in clearance with 50%; 40%; 30%; 20% cardiac reduction respectively. Numbers are calculated percentage change (decrease) in mean clearance (CL) following corresponding reduced cardiac output (CO) as described in Supplemental Materials, Section 2) [27-38].”
Same p6L180ff.
Reply 27: The quoted text was intended as a legend and not part of the main text. This has been made clearer in this instance (see P11L177) and throughout the manuscript. The editing is provided in line with the standards of the journal.
Clarification of populations (parameters & simulations)
The information on the differences between the different age populations is not sufficient.
Comment 28:
- Clarification of population simulations.
The data on the virtual populations is not provided and not clearly described. As I understand it 5 virtual populations (neonate, infant, child, adolescent, adult) of respectively 200 subjects were simulated.
These populations are at the core of this work and should be provided in the supplement. I.e. add a table with all parameter changes for the 200 x 5 subjects simulated. This should include the physiological changes (body weight, organ weights, blood flows, and additional parameter changes (e.g. HLM).
The base model in combination with these table of parameter changes should allow to simulate the 5 populations.
- The definition of the groups is missing from methods, i.e. how are neonate, infant, child, adolescent, adult defined (age cutoffs).
- What are age dependencies in the model for the different groups, i.e. which parameters change with age in the model? Which parameters change between the different groups. Some information exists in Michelet2018 (Table 8) but it is unclear which additional age dependencies are in the model. E.g. it is known that hepatic blood flow changes with age (which will be key determinant of hepatic clearance), but it is unclear if the hepatic blood flow changes with age in the model. Referencing the original publication which is not well documented is not sufficient, but all information necessary to understand/reproduce the current work must be provided. A table should be provided with the detailed changes, only stating “the ‘Paediatric’ virtual population of the Simcyp simulator which contains the physiological changes in children and the ontogeny patterns of CYP2B6, 2C9, UGT1A9” is not sufficient.
=> An overview table of all physiological changes between populations should be provided.
Reply 28 (The following response addresses all the issues raised in comment 28 above)
As explained in reply 1, all population models within Simcyp have been previously reported and published. We do not believe it is the remit of authors to report this, since as already explained, this research was not focused on model development.
Goodness of fit plots
Comment 29:
Provide goodness of fit plots:
- See Rüdesheim2020 Figure 3 for an example
- The observed vs. predicted data should be depicted (use color to show the different groups in the plot).
- Also provide the goodness of fit plot for the predicted parameters, i.e. for CL and t1/2 (data shown in Table S3). This is the main model evaluation and belongs with the other goodness of fit plot in the main manuscript (see for an example Figure 5 in Rüdesheim 2020).
Reply 29: The authors would like to bring the reviewers attention to the validation strategies the authors described will be used for evaluation of the model (see supplementary material P1L17-27). Since, there is no general consensus on the how validations data should be presented, we believe the approach we have chosen is acceptable and has been used previously used by other researchers [8, 9].Also, in contrast to the paper the reviewer refers to (Rudesheim et al 2020) which was focused on model development, this current research aimed at simulation of a ‘what if’ scenario, we therefore believe the validation steps we have reported are sufficient.
Dosing schema
Comment 30:
I don’t understand the used dosing schema.
“Following validation of the optimized propofol model, the Morse et al. dosing model [39] was simulated in neonates, infants, children, and adolescents, while the Roberts et al. dosing model [40] was implemented in adults …”.
Morse does to our knowledge not provide information on adolescents, but only up to 1-2 years? Why/how is morse applied to children and adolescents?
Reply 30: The authors would like to bring the reviewer attention to the Table 3 in Morse et al paper which reported a dosing for 1 – 3 years. This age classification falls within the the International Conference on Harmonisation(ICH) E11 classification’s [10] classification of children. Also, the author will like to bring the reviewer’s attention to the legend of Table S4 where the authors mentions that the children dose was used in adolescents.
Dosing optimization
Comment 31a:
- The dosing optimization should be implemented more thoroughly. I.e. the actual dose reduction per age group should be actually optimized and the value determined which achieves the target value 3 μg/mL @ 2hrs for the mean/median of the population should be reported. Alternatively the percentage of subjects in the TR in the population could be maximized.
Reply 31a: The actual dose used were reported (see Table S4 supplementary material).
Comment 31b:
I would expect to see optimized values which are not exactly 15% but 12.3% for instance. The recommendation can be a 15% reduction, but the optimization has to first determine the optimal value for dose reduction.
Reply 31b: The authors regret this could not be implemented in this study but we expect that the percentage reductions studied will provide the readers a broad enough insight on what might happen within and around the range of CO reductions studied.
Validation data missing
Comment 32:
- not a single datapoint is shown for reduced cardiac output and compared with the model. As I understand this work it uses an existing model and simulates CO changes. So none of the predictions in this work have been validated. A systematic literature research should be performed on propofol pharmacokinetics with reduced CO (e.g. in heart disease) and changes in the pharmacokinetics parameters compared with model predictions. References are misleading, unclear importance of CO reduction.
Reply 32:
The authors argue that based on evidence from Leslie et al 1995 who demonstrated significant alteration in propofol PK [11]; several reported instances of reduced CO in hypothermic condition [12, 13]; and reported evidence that propofol is a high extraction drug that is sensitive to CO changes[14], there is a high chance that reduced CO correlates with propofol PK. We accept that our finding are theoretical at the moment and are subject to clinical validation and apart from the fact that our findings may provide insight into the PK of propofol under certain cardiac conditions, it could also serves as a framework to guide clinical studies involving propofol in patients with compromised cardiac outputs in the future – we consider this as one of the strengths of this research.
We have added extra point to provide further clarification and to better explain our perspective to the results we have obtained (See P14L301-311)
Comment 33:
- Some of the references do not state what the sentences suggest. E.g.
p9L247
“CO is documented to affect propofol clearance in adults [3,6,10]”. I don’t think reference [3] includes CO data, [6] is a review (original work should be cited showing the stated fact), as is [10] (reference original research). You even state yourself that this study has nothing to do with CO: “However, as this study did not measure the CO in the hypothermic subjects, it is difficult to link the reduced CO to propofol clearance [3].”
I did not see any data on the effect of CO on propofol pharmacokinetics and are not very convinced that there is a strong effect. E.g. Bienert2020 states: “On the other hand, in a pilot study by Peeters et al. [4] no significant relationship between measured CO and propofol clearance in ICU patients was observed. Therefore, it is still an open question as to whether CO may really by useful to predict the concentrations of propofol or fentanyl.” and “The inclusion of CO into the model led to a small decrease (less than about 20%) in inter-individual variability of PK parameters as presented in Table 2S. It indicates rather limited clinical applicability of the final model in predicting drug dosing in comparison to a simple model (without CO effect on PK/ PD parameters).”
Also this work only showed effects for very large CO changes > 30%: “Applying this target, CO reduction (up to 30%) did not result in mean plasma concentrations outside of this target range across all age groups (Figure 3).” This are key sentences for the manuscript, the references have to be correct and key aspects such as the importance of CO reduction for propofol pharmacokinetics have to be clearly and critically discussed.
Reply 33: The impact of CO on propofol PK is debatable and the authors argue that though Biernet did not identify observed CO reduction as a significant covariate impacting the propofol PK in patient with only about 20% alteration to variability, Biernet found significant model predicted CO correlations between CO and propofol PK. Biernet et al explained why the measured CO may not have improved variability in propofol PK parameters as quoted “The less significant impact of the measured CO on PK of studied drugs can be related to the fact that (a) the measured CO only approximates organ blood flow and (b) it is a measured variable that is stochastically related to the true value of liver and tissue blood flow of each subject.”
In statement and the reply 32 provides the basis for the perspective to this results that the authors have. We have added an extra paragraph to make our point clearer. (See P14L301-311)
Minor issues
Comment 34
- P2L57
You state “The clearance of propofol - a high extraction drug is sensitive to CO changes in adults [8].” The reference [8] does not contain any data/information on whether the clearance of propofol is sensitive to CO changes! (it only studies the contribution of various sites to the clearance of propofol).
Provide a correct reference supporting the statement or reformulate in a more general statement along the line of: high extraction drugs are usually affected by blood flow/cardiac output. Because propofol is a high extraction drug as shown in [8] it can be expected that changes in CO would affect the pharmacokinetics of propofol”. Perhaps you mean reference [20] ?!
Reply 34: The authors have added new reference (See P2L57)
Comment 35:
- P2L77
“Physiologically based pharmacokinetic (PBPK) modeling is a non-invasive and reliable approach” -> remove “and reliable”. The reliability depends strongly on the used model, data, application and experience of the modeler. Such a general statement that PBPK modeling is reliable is not arranted.
Reply 35: Rather than remove the term ‘and reliable’ we have included extra text to address the authors comment. The extra text included is italicized in the quoted below
“Physiologically based pharmacokinetic (PBPK) modeling is a non-invasive and when implemented appropriately, is a reliable approach...” (see P2L77)
Comment 36:
- P2L79
“enabling athe ssessment” -> typo: “enabling the assessment”
Reply 36: The typo has been corrected. (see P2L79)
Comment 37:
- P2L79
“impact of altered (path)physiology” -> typo: (patho)physiology; altered pathophysiology does not make sense, meaning changes of the already changed physiology in disease; Change to altered physiology.
Reply 37: The authors note this typo and have instead of the reviewer’s suggested changed have corrected “(path)physiology” to (patho)physiology and removed the term ‘altered’. (see P2L80)
Comment 38:
- P2L88
“and clearance is affected by alterations in CO in adults”; add reference for the statement which clearly shows this fact, or if this is just inferred from the high extraction ratio clearly state this. E.g. “with clearance therefore expected to being affected by CO”.
This assumption/statement is at the core of this manuscript but no clear reference/data is provided that CO is affecting the propofol pharmacokinetics!
Reply 38: The authors have made this correction. Please (see P2L88-89)
Comment 39:
- P3L91
“assess the impact of reduced CO in low CO settings”. This statement is confusing/wrong. It sounds as the reduction of CO of already reduced CO values was studied, but only the reduced CO of normal CO values was studied. Remove “in low CO settings”.
Reply 39: The authors have corrected this. (see P2L92)
Comment 40:
- P3L104
“and peer-reviewed propofol model (Michelet et al.) was optimized and further validated [22].” Unclear how the model was further validated. Validation requires the comparison of model simulations (not used in optimization) with data. I did not see any such comparisons in the manuscript. Remove “was further validated”.
Reply 40: The authors included this wording because they deemed it necessary to validate the optimised model to ensure the model used for the current research was predictive of data reported in observed studies. This was also needed because of the several upgrades of Simcyp software since the original model was developed.
Comment 41:
- p3L118f
Provide information on how CO affects blood flow in the model I assume that the model uses Q_H = fQ_H * CO and Q_R = fQ_R * CO, i.e. a fraction of the CO is going as venous blood to the organs. As a consequence a 10% reduction reduces also the tissue blood flows by 10%. How CO and Q_H and Q_R are modeled should be stated, because this is key to understanding the effect of reduced CO in the model predictions.
Reply 41: The authors have added more information. (see P3L118-121)
Comment 42
- p3L121
References are positioned incorrectly “Using the Morse and Robert models, the impact of CO reduction on systemic propofol clearance was determined by calculating the percentage relative change under reduced CO conditions [39,40]” => “Using the Morse and Robert models [39, 40], the impact of CO reduction on systemic propofol clearance was determined by calculating the percentage relative change under reduced CO conditions”. This reads as if the references are for the calculation of the percentage change.
Reply 42: The authors have corrected this. (see P3L125)
Comment 43:
- target plasma concentration of 3
Reply 43: The authors have corrected this. (see P3L138)
Comment 44:
- (80-125%) μ g/mL in children [39] and adults [40] respectively
Reply 44: The authors have corrected this. (see P3L138)
Comment 45:
- p4L152
“ratio target (0.7). .” remove second dot
Reply 45: The authors have corrected this. (see P3L161)
Comment 46:
- p4L153
The following text reads very complicated and not very clear:
“In Figure 2, the relative change in clearance following CO reductions increased with age with the least impact of CO reduction in neonates across all CO reduction scenarios. The greatest magnitude of change occurred between neonates and infants, suggesting the greatest developmental changes impacting propofol clearance in the first year of life. Fig- ure 2 shows that higher percentage CO reductions resulted in greater relative percentage reductions. The impact of CO reductions across age groups was greatest when CO was reduced by 50% with relatively lower percentage clearance reduction across age group when CO reduction was limited to 20%.
Reply 46: The authors have corrected this. (see P5L164-175)
Comment 47:
- p6L190
“Figure 4 shows that dose reduction of 40 and 50% reduced CO conditions resulted in predicted plasma concentrations within the target range in all age groups.” Sentences not understandable, reformulate.
Reply 47: The authors have changed the text referring to Figure 4. (see P10L212-219)
Comment 48:
- p9L238
“[8]. ]The” incorrect brackets
Reply 48: The authors have corrected this. (see P14L273)
Comment 49:
- p9L253
“This confirms the midazolam pattern, with an EH low extraction (0.02) pattern at birth and intermediate extraction (6.0) during adult- hood [44].” The presented results do not confirm the midazolam pattern, they just show a similar pattern then observed for midazolam. Rephrase.
Reply 49a: The authors have corrected this. (see P14L288)
p9L266
“in clinical conditions such neo- natal asphyxia undergoing” Word missing
Reply 49b: The authors have corrected this. (see P14L312)

Reviewer 2 Report
Well written manuscript with interesting results (and inline with the expectations). Only a few point for improvement, it did not get clear to me how the propofol infusion is administered, the rationale for the concentration target and how the CO input is done within Simcyp.

Author Response
Reviewer: 2
Comment 1
Throughout the manuscript it does not get clear to me whether it is an iv bolus infusion or whether dose are altered to get the target at C2h. Please clarify this in the different sections.
Response 1:
The authors have made this clearer by adding extra text (Please see P1L23 and P3L17)
Comment 2
Please explain where this target comes from..
Response 2
The authors have added extra text to explain this. Please see section 2.3 (P3L138)
Comment 3
It does not get clear to me how this was implemented in Symcyp. Please expand.
Response 3
The authors have made this clearer by adding extra text (Please see P3L130-131).
Comment 4:
Two Section 2.2.
Response 4:
This has been corrected. (Please see P3L137 and 142).
Comment 5:
Please explain again where this target comes from.
Response 5:
The authors have added extra text to explain this. Please see section 2.3 (P3L138)
Comment 6:
Please re-iterate the infusion duration
Reply 6: The authors have added an extra text (P3L142)
Comment 7:
It is not clear why the conc. increase again after time. How can these spikes be explained? Please expand a bit in the figure footer/header. Please re-iterate that this is following an iv bolus
Reply 7: The spikes appear so for two reasons. Firstly, the dosing models require administration of doses at specific times (which appear like the spikes in the figure). Secondly, the spikes are rather obvious or pronounced because of the log scale of the y-axis; in practice, this is not likely to have any significant consequences.
Comment 8:
To be deleted
Reply 8: This has been deleted (see P14L273)
Comment 9:
Subscript H to be consistent
Reply 9: The authors have corrected this (see P13L285)

Reviewer 3 Report
The manuscript was structured well and provides a good opinion supporting the study endpoints.
There is a point that the authors need to consider for the real clinical situation.
1. The authors simulated the situation regarding reduction of cardiac output upto 50%.
However, in the real cinical situation, low cardiac output exceeding 50% of reduction is frequently observed. As a result, if the expectation or opinion about severly reduced cardiac output (>50%) would be porvided, the clinical application could be greater.
Author Response
Reviewer: 3
Comment 1:
The authors simulated the situation regarding reduction of cardiac output upto 50%.
However, in the real clinical situation, low cardiac output exceeding 50% of reduction is frequently observed. As a result, if the expectation or opinion about severely reduced cardiac output (>50%) would be provided, the clinical application could be greater.
Reply 1: The authors would like the reviewer to note that first, we have chosen this range of percentage reductions to give a broad overview of effect of CO reduction. Readers may expect that in >50% reduction scenarios, at minimum, the intervention applied may be similar to that demonstrated in the case of 50% reduction in this study. Secondly, when patients have >50% reduced CO, it may not be clinical viable to induce anesthesia with propofol. We believe and expect that in such scenarios, other compounds like eg ketamine will be considered. This is also reflected in the propofol SmPC (‘Propofol should not be administered in patients with advanced cardiac failure or other severe myocardial disease except with extreme caution and intensive monitoring’)
The authors would like to thank the reviewers for their time in suggesting improvements to our manuscript and for the Editor for consideration of our body of research.
Olusola Olafuyi
Mohammed Abassi
Robin Michelet
Karel Allegaert

Round 2
Author Response
Thanks so much for the detailed revision. Most of the issues were resolved and the manuscript improved substantially. The updated/added plots are great and provide a lot of additional information.
Thank you, we hereby understand that all issues, except model availability, have been addressed, and that the reviewer consider these as solved.
Only issue remaining is the model availability.
Major issue: Model availability
Unfortunately, I don’t understand why it is not possible to fulfill the minimal requirement of making the model available. It is most likely impossible to obtain the actual model equations due to the black-box character of SIMCYP, but providing the actual model file in the supplement is one minute of work. Why is it not possible to provide the SIMCYP file of the model in the supplement?
By providing the model file minimal model consistency as well as basic model behavior can be validated. It is the task of a reviewer to ensure that the main results are valid. Especially for computational models and analysis this can easily be performed and reviewers are encouraged to do so. As part of a manuscript “the minimal dataset that supports the central findings of a published study” should be made available. In case of an adapted computational model this is the actual computational model.
Response
We are not aware of the background of the reviewer, but data access within a GDPR setting quite commonly needs a data sharing agreement, or - at least - this is how my employer addressed these issues, irrespective on my personal opinion on this. This means that data can be shared upon reasonable request, and following a detailed data sharing document. This means we – or at least me as corresponding author to KU Leuven - cannot share the raw data on the neonatal population (the access to these raw data is discussed in the paper). We have rediscussed this within the authorship team, and what we can release has been provided as the adult, paediatric and a CO simcyp model file.

Round 3
Reviewer 1 Report
Thanks so much for the additional supplements.
Table 1 is not layouted correctly and not readable in the PDF. This should be fixed in typesetting.